# Antidiabetic, Antioxidative and Antihyperlipidemic Effects of Strawberry Fruit Extract in Alloxan-Induced Diabetic Rats

**DOI:** 10.3390/foods12152911

**Published:** 2023-07-31

**Authors:** Iftikhar Younis Mallhi, Muhammad Sohaib, Azmat Ullah Khan, Imtiaz Rabbani

**Affiliations:** 1Department of Food Science & Human Nutrition, University of Veterinary & Animal Sciences, Lahore 54000, Pakistan; 2Department of Physiology, University of Veterinary and Animal Sciences, Lahore 54000, Pakistan

**Keywords:** strawberry extract, antidiabetic, flavonol, health promotion, hypolipidemic, histopathology

## Abstract

Strawberry (*Fragaria × ananassa*) is one of the accomplished sources of bioactive compounds, including anthocyanin, phenolic acids, flavonols, ellagitannins, and a diverse range of minerals and vitamins that can help to boost human health. This study was carried out to explore the antidiabetic, antioxidative and antihyperlipidemic potential of strawberry extracts against alloxan-induced (100 mg/kg body weight) diabetic rats. Accordingly, rats were categorized into six groups including control (G_0_), positive control (G_1_), treatment groups (G_2_, G_3_, and G_4_) given strawberry extract at 250, 500, and 750 mg/kg of body weight, respectively, and G_5_ provided metformin @70 mg/kg BW for 28 days with ad libitum diet. At the trial termination, the rats were sacrificed and were subjected to analysis including body weight, blood glucose level and glycemic indicators, antioxidant parameters, lipid profile, renal function test (RFT), liver function test (LFT) and histopathology for pancreatic tissues. The results indicated that treatment of diabetic rats with strawberry extract at 500 mg/kg body weight (BW) resulted in significant reductions in blood glucose level, serum urea, and creatinine as well as significant increases in body weight, insulin activity, and protein levels. In addition, the diabetic rats that did not receive strawberry extract (control) exhibited an increase in plasma glucose, urea, uric acid, creatinine, and a decrease in body weight and insulin levels. Briefly, it is reported that strawberry fruit extracts reduced blood sugar levels, possess hypolipidemic potential, and helped to maintain antioxidant levels in alloxan-induced diabetic rats.

## 1. Introduction

The metabolic abnormalities of diabetes mellitus have a significant link to mortality and morbidity. It is estimated that approximately 537 million adults (20–79 years of age) around the globe live with diabetes according to the latest statistics provided by the International Diabetes Federation (IDF). It has been estimated that approximately 783 million diabetics will live in the world by 2045, compared with 643 million in 2030. According to IDF statistics, three out of four adults with diabetes are living in developing countries. In 2019, approximately, 4.2 million deaths occurred due to diabetes, and the number of deaths caused by diabetes is increasing steadily every year. Health expenditures associated with diabetes totaled at least USD 966 billion, or 9% of total adult health expenditures. According to the International Diabetes Federation (IDF), Pakistan has a high load of diabetes; an expected 14 million people are living with diabetes in 2019, and the prevalence of diabetes in Pakistan is increasing due to a number of factors, i.e., population growth, urbanization, sedentary lifestyle, and poor dietary choices. Moreover, Pakistan has the 3rd largest number of diabetic patients in the world, after China and India [1].

Diabetes can cause a number of complications, including insulin resistance, hyperglycemia, oxidative stress, neuropathy, retinopathy, nephropathy, stroke, cardiovascular disease, gum infection, amputations, polydipsia, polyphagia, polyuria, muscle weakness, weight loss, hyperglycemia, and glycosuria. The literature reported that compared to pharmaceutical drugs such as insulin, sulfonylurea, and thiazolidinediones, natural remedies are safer in some ways. In the past few decades, interest in natural hypoglycemic agents has increased due to their fewer toxic side effects than synthetic hypoglycemic agents. It has been shown that strawberry fruit can be beneficial to people with type 2 diabetes mellitus (T2DM), owing to its rich nutritional and phytochemical composition. The strawberry fruit can be used in a variety of ways to treat type 2 diabetes mellitus (T2DM), improve insulin sensitivity, reduce blood vessel inflammation, and improve endothelial function by consuming fresh, frozen, extracted, and even powdered extract, whereas fresh fruit has better nutritional value compared to other counterparts. Studies have shown that strawberries may impact type 2 diabetes mellitus biomarkers directly in animal models, owing to the nutritional composition and phytochemical profile of strawberry fruit. Additionally, one of the studies reported that composition and function of intestinal microbiomas of T2DM-induced mice were significantly altered when strawberries were supplemented with nutritional doses [2].

Strawberry fruit contains health-promoting compounds such as phenols, flavonoids, phenolic acids, and anthocyanins, as well as antioxidants such as ascorbic acid, ellagitannins, and anthocyanins, which enhance oxidative stability of living organisms, leading to improved health. Strawberry extract also contains folate, a water-soluble vitamin B derivative (tetrahydrofolic acid) that is also used as a supplement in food fortification programs around the world. Furthermore, strawberry phytochemicals may contribute in quality, sensory, and organic properties of the fresh fruit and strawberry-based processed fruit products. Strawberry polyphenols mainly consist of anthocyanins, and it has recently been reported that strawberries are one of 100 richest sources of dietary polyphenols consumed by people in the world. Different research studies have shown that anthocyanin profiles differ among strawberry varieties due to genetic background, ripeness level, and storage conditions after harvest of the fruit [3].

It has been reported that strawberry fruits have a high concentration of bioactive compounds that are beneficial for consumer health. Studies have documented that strawberry shows effects on cardiovascular disease and diabetic complications with an ultimate positive impact on human health. Free radicals in the body are significantly inhibited by strawberry juice [4]. One of the research groups examined the effects of freeze-dried strawberry supplementation on glycemic control, lipid profile, and serum antioxidant levels in diabetic patients. They randomly divided 36 subjects (23 females) with T2DM into groups, with mean BMI and age of 28 and 52 years, respectively. In a six-week randomized double-blind controlled trial, participants were given two cups of freeze-dried strawberry beverages, followed by measurements including anthropometrics, dietary intake, HbA1c, anti-oxidants, C-reactive protein, and malondialdehyde (MDA) levels at baseline and post-intervention. The results reported that at the sixth week, strawberry supplementation significantly reduced C-reactive protein (2.4 vs. 2.03 mg/L, *p* < 0.05) and lipid peroxidation in the form of MDA (3.3 vs. 2.7 nmol/mL, *p* < 0.05) levels. Furthermore, supplementation led to a significant improvement in total antioxidant status (1.44 vs. 1.26 mmol/L, *p* < 0.01) as well as a reduction in HbA1c (–5.7%, *p* < 0.05). The results also showed positive impact on glycemic control, antioxidant status, lipid peroxidation, and inflammation in patients with type 2 diabetes. It is well known that freeze-dried berry-based products contain antioxidants and have a low glycemic index, which helps alleviate metabolic complications associated with type 2 diabetes mellitus [5].

Despite numerous reports on strawberry health benefits, no conclusive studies have examined the anti-diabetic, antioxidative and antihyperlipidemic potential of strawberry against experimentally induced diabetes in vivo. Therefore, the present study was carried out exploring different strawberry (*Fragaria* × *ananassa*) extracts for their antidiabetic, antioxidant and hypoglycemic effects using rat animal models.

## 2. Materials and Methods

### 2.1. Chemicals and Reagents

The chemicals used to carry out the trial were purchased from Sigma-Aldrich (St. Louis, MO, USA). Analytical kits for cholesterol FS 10 (cat. no. 113009910923), Glucose GOD FS 10 (cat. no. 125009910923), alanine transaminase (ALT) (GPT) FS (IFCC mod.) (cat. no. 127019910920), creatinine FS (cat. no. 117119910920), high-density lipoprotein-c immune FS (cat. no. 135219910920), triglycerides FS (cat. no. 157109910923), alkaline phosphatase (cat. no. 104419910920), and low-density lipoprotein cholesterol Selectra (cat. no. 141219910921) were purchased from Response^®^-Diagnostic Systems, Holzheim–Germany and rat feed ingredients were purchased from the local market. Alloxan monohydrate and metformin (MET, positive drug) were purchased from Sigma Chemical (St. Louis, MO, USA) and a local pharmacy in Pakistan, respectively. The rat insulin enzyme linked immunosorbent assay ELISA kit, superoxide dismutase (SOD) and glutathione peroxidase (GPx) were performed using commercially available kits (Randox, UK). Strawberries for the trial were purchased from local markets and farms in Sharqpur, Lahore.

### 2.2. Strawberry Sampling and Extract Preparation

For the execution of research, fresh strawberry samples were collected from farms in Sharqpur, Punjab, Pakistan. After sample collection (400 g), strawberry samples were cut into pieces, air-dried under shade (30 °C) for 10 days, blended, and then grinded to obtain a refined powder [5]. The strawberry powder was mixed with 95% methanol and stirred with a magnetic stirrer overnight at room temperature. Afterward, by using Whatman No. 1 filter paper, residues were removed, and filtrate was collected that was concentrated under vacuum using a rotary evaporator at a temperature of 30 °C for the extract. Until the entire methanol had evaporated, the procedure continued, yielding 36.8 g of crude extract (9.2%) that was further used for the animal model study.

### 2.3. Animal’s Procurement, Housing and Handling

Accordingly, 48 normal glycemic male adult rats weighing 150 ± 5 g aged 6 weeks were purchased from the University of Veterinary and Animal Sciences (UVAS), Lahore, and acclimated with basal diet for the period of 12 days in the shed of the animal room facility of the university. The stainless steel cages were used to house the animals, and wood litter bedding was used as floor material that was changed every week to maintain hygiene and to protect them from disease. In addition, a temperature of 24 ± 2 °C was maintained under the cycle of 12 h (12 a.m.–12 p.m.) in an environmentally controlled room for rats. The fresh water and a standard rat diet (5.8% sunflower oil, 20% dextrose, 22.5% casein, 5.8% sunflower oil, 40.7% maize starch, 9.7% minerals, and 1.3% vitamins) were provided to rats ad libitum. Body weight (BW) was measured on the first (baseline) and last day (termination) of the research trial. Before conducting the trial, the university animal ethical committee approved all methods and procedures required for the trial and analysis as well as issued an ethical certificate with detail (issue no. DR/ 575; dated: 28 September 2022).

### 2.4. Experimental Design

In the research study, the rats were divided into six different groups, i.e., G_0_: negative control that received 0.9% NaCl solution; G_1_: positive control, received a single dose of alloxan monohydrate 100 mg/kg; G_2_, G_3_, and G_4_ are diabetic groups that received alloxan (Single dose) and dietary intervention of strawberry extract at 250, 500, 750 mg/kg of body weight, respectively, for 28 day; G**_5_** rats were also diabetic and given metformin dose 70 mg/kg for 28 days. All the groups received ad libitum miller bander basal feed during this trial. For diabetes induction, rats were kept on overnight fasting conditions after diet feeding for two weeks to acclimatize them. Following that, 100 mg/Kg BW of alloxan monohydrate (Sigma, USA) was injected intraperitoneally in 0.5 mL of saline solution. Rats were given a 20% glucose solution after 6 h of alloxan administration to prevent hypoglycemia. A fasting blood sample was collected from the tip of the tail three days after alloxan injection, and glycemia was measured (ACCU CHEK, Corydon, IN, USA). Study participants were only considered diabetic if their fasting blood glucose level exceeded 250 mg/dL. After induction of diabetes, the experiment lasted for 28 days. After the 28-day experimental period, all rats were euthanized by following the standard protocols described in [6], and blood sampling was performed using heart puncture technique.

### 2.5. Monitoring Parameter

An electronic balance (Model: UX 420H, Company: Shimadzu-Japan) was used to monitor the change in body weight of rats on the 1st (initiation of trial) and 28th (termination) day along with other parameters after diabetes induction in rats.

### 2.6. Blood Chemistry Analysis

After blood sampling, serum was centrifuged and collected for analysis that was performed using blood chemistry analyzer (Response^®^ 910, Diagnostic System-SIEMENS, Munich, Germany) for the following parameters; low density lipoproteins (LDL), high density lipoproteins (HDL), triglycerides (TG), total cholesterol (TC), alkaline phosphates (ALP), alanine amino transferase (ALT), creatinine, albumin, and total protein. The serum chemistry data of rats provided strawberry as dietary intervention were analyzed using software version 2.2.3.3.

### 2.7. Antioxidant Parameters

Photometry was used to measure SOD activity in erythrocytes of rats given different levels of strawberry extract as described by [7] using the RANSOD kit (RANDOX Laboratories, Ltd., Crumlin, UK). The superoxide dismutase (SOD) activities in the rat blood samples were calculated per mL of international units and expressed as (U/mL).

Further, to determine the activity of GPx in erythrocytes, the photometry method was used according to [8] using the RANSEL kit (RANDOX Laboratories, Ltd., Crumlin, UK). The GPx value was calculated per mL of international units and expressed as (U/mL).

### 2.8. Fasting Serum Glucose

Two measurements of fasting blood sugar were measured for the rats provided the dietary strawberry fruit extract. The first blood fasting glucose was measured after diabetes was induced, and the second reading was measured at termination (28 days) of the research trial. For measuring blood glucose, animal tails were sterilized with 10% alcohol, cut with scissors, and blood was allowed to touch a test strip by following the guidelines as described by [9]. Afterward, within 5 s, a direct blood glucose reading was noted in mg/dL.

### 2.9. Histopathology Study

After the dissection of rats, a portion of pancreatic tissue was separated and stored in 10% neutral buffered formalin and processed. The histopathological samples were processed in the following manner, including dehydration, clearing and infiltration. Tissues of 0.4 µm thickness were cut after preparation of blocks. Hematoxylin and eosin (H&E) staining of tissue sections was carried out as per the guidelines described by [10]. Photomicrographs were taken after sections were examined under a light microscope, which were further utilized for the impact assessment of treatment groups compared to control.

### 2.10. Statistical Analysis

All obtained data were analyzed through SPSS version 20. The results were expressed as mean ± SD. One-way analysis of variance (ANOVA) was performed to find the significant differences. Post hoc test, i.e., Duncan’s multiple range test (DMRt), was performed on the significant differences among parameters to assess the differences (*p* < 0.05) between the groups by following the guidelines of [11].

## 3. Results

### 3.1. Effect of Strawberry Extract on Body Weight, Serum Glucose and Insulin Level

The results (Table 1) showed significant differences among body weight and blood glucose level from the 1st to 28th day of diabetes induction in rats as well as (Figure 1) the insulin level of all six groups (G_0_, G_1_, G_2_, G_3_, G_4_ and G_5_). Diabetes reduced the body weight of the rats, and the dietary intervention of strawberry extract significantly increased the body weight of diabetic rats compared to control. Means for weight gain of the G_0_ (negative control, non-diabetic) group had an initial mean of 155.04 ± 7.44 and after 28 days of 205.89 ± 7.56. Similarly, group G_1_ (positive control, diabetic with standard diet) reported a reduction in body weight from 158.42 ± 6.53 to 137.51 ± 4.37. Further, diabetic groups G_2_, G_3_ and G_4_ treated with strawberry extract 250, 500 and 750 mg/kg of body weight showed positive effects compared to control group. Group G_4_ (diabetic + orally induced 500 mg/kg of strawberry extract) reported reduction in body weight from 157.01 ± 7.44 to 148.84 ± 4.56, which is the lowest body weight reduction as compared to the diabetic control group. Similarly in group G_5_ (diabetic + treated with metformin) reduced in body weight was observed whereas, metformin did not control weight compared to strawberry fruit extract. The blood glucose level of rats provided the strawberry extract was also significantly affected (Table 1). Means for the G_0_ negative control (nondiabetic + standard diet) were documented as 87.23 ± 3.06 and 88.3 ± 4.50 on the 1st and 28th days, respectively. The G_1_ group (positive control with diabetes with standard diet) showed a high level of glucose at 409.65 ± 18.87 and 474.85 ± 20.01 before and after, respectively. Results also reported that the treated groups G_2_, G_3_ and G_4_ sustained a decrease in blood glucose level after 28 days of treatment compared to control. These groups were orally administrated strawberry extract 250 mg/kg of body weight for G_2_, 500 mg/kg for G_3_ and 750 mg/kg of body weight for G_4_. Likewise, group G4 reported the highest decrease in blood glucose level from 402.41 ± 19.51 to 231.41± 9.22. The diabetic rats (G_5_) treated with metformin has reduced blood glucose from 397.34 ± 16.03 to 269.97± 14.53, which is less effective as compared to strawberry extract [12]. The results documented that diabetic control G_1_ had decreased insulin level. The insulin level of the diabetic control group (G_1_) was found to be decreased in comparison with that of the non-diabetic control group (G_0_), as reported in (Figure 1).

### 3.2. Effect of Strawberry Extract on Liver Function

As shown in Table 2, strawberry fruit extract treatment reverted to normal for parameters of liver that were elevated/decreased in the diabetic group of rats. The bilirubin value in the control group G_0_ (non-diabetic + standard diet) was reported to be 0.24 ± 0.02, and the value for the rats in G**_1_** (positive control with diabetes + standard diet) reported a value 0.16 ± 0.03 and reverted to normal values in the treatment groups, i.e., G_2_, G_3_, G_4_, and G_5_. ALT, AST, and ALP levels were elevated in diabetic rats according to normal rats (G0). In control group G_0_, ALT, AST, and ALP values were 43.31 ± 1.23, 74.50 ± 1.98, and 152.00 ± 7.81, respectively. After diabetes induction, the values were elevated, as ALT was recorded at 56.44 ± 1.04, AST at 219.71 ± 0.78, and ALP at 347.33 ± 4.51, and the results documented that strawberry extract significantly decreased the value of ALT, AST, and ALP to 45.96 ± 0.75, 199.47 ± 1.90, and 164.54 ± 1.94, respectively, in G3. In the diabetic group (G_5_) that was treated with metformin, the documented values were 51.45 ± 1.19, 214.22 ± 0.99, and 246.00 ± 4.36 for ALT, AST, and ALP, respectively. Compared to the control, the diabetic rats showed a significant decrease in levels of total protein, albumin, and globulin, whereas treatment groups G**_2_**, G**_3_**, and G**_4_** showed that total protein, albumin, and globulin levels were higher than G_1_ (positive control) as reported in (Table 2).

### 3.3. Effect of Strawberry Extract on Antioxidant Parameters

Antioxidants, GPx and SOD play a crucial role in fighting against oxidative stress that ultimately improves the health status of the living organism. It has been observed that glutathione peroxidase (GP_X_) of the diabetic group (G_1_) was significantly low (0.82 ± 0.21) compared to the non-diabetic group (G_0_), at 2.83 ± 0.21. The plasma GP_X_ activity of the treatment group (G_4_) was significantly high (1.72 ± 0.34) compared to G_2_ (0.95 ± 0.01) and G_3_ (1.54 ± 0.21). Similarly, the plasma superoxide dismutase (SOD) activity of diabetic rats in G_1_ showed (2.54 ± 0.26) a significant decrease compared to non-diabetic group G_0_ at 7.53 ± 0.31. The other treatment groups, G_2_ at 4.13 ± 0.18, G_3_ at 4.74 ± 0.23 and G_4_ at 5.49 ± 0.32, showed positive response against strawberry fruit extract as reported in (Table 3).

### 3.4. Effect of Strawberry Extract on Renal Functions

The findings of different renal markers such as serum creatinine and urea of rats provided with the strawberry fruit extract along with control groups are presented in Figure 2 and Figure 3. The control (G_0_) group mean values for serum creatinine and serum urea were reported as 0.59 and 21.59 mg/dL, respectively, and for the diabetic control group (G_1_), the values were 1.07 and 43.69 mg/dL, respectively. The treated animal group exhibiting recoupment toward the control, and groups G_3_ and G_4_ exhibited close reversal to the normal rat group (G_0_).

### 3.5. Effect of Strawberry Extract on Plasma Lipid Profile of Rats

The results (Table 4) indicated that low-density lipoprotein (LDL), triglycerides (TG), total cholesterol (TC), and serum high-density lipoprotein (HDL) were reduced in diabetic rats compared to non-diabetic rats. The total cholesterol level in (G_0_) rats was 71.00 ± 4.03, and after diabetes induction, the values increased to 75.21 ± 3.02 mg/dL. Serum triglycerides (TG) values in G_0_ were 103.33 ± 5.15, and were elevated in G_1_ 135.33 ± 6.53. Serum low-density lipoprotein (LDL) values in G_0_ were 14.41 ± 0.27 and were elevated in G_1_ as 21.67 ± 0.24. Serum HDL values reduced from 24.23 ± 1.30 to 15.21 ± 0.59 in the diabetic rats (G_1_) group. All these values were close to normal in the treated groups G_2_, G_3_ and G_4_. The diabetic groups that were treated with 250, 500 and 750 mg/kg of BW strawberry extract resulted in a return of normal levels of high-density lipoprotein (HDL) cholesterol and very low-density lipoprotein (VLDL) cholesterol, which were close to the levels in the control non-diabetic rats.

### 3.6. Effect of Strawberry Extract on the Pancreas of Rats through Histopathology

Figure 4 shows the H&E-stained sections of the paraffin-embedded pancreatic tissues of rats in the control and experimental groups (1st and 28th day of diabetes induction). The histopathology examination of the pancreases of rats showed that the pancreas reversed to normal after destruction and microscopic examination.

## 4. Discussion

The present data demonstrate the effect of consumption of strawberry on major health parameters, including body weight, fasting serum glucose concentration, serum insulin, bilirubin total, liver enzymes, total protein, albumin and globulin, antioxidant parameters, plasma lipid profile, and effect of strawberry extract on the pancreas of rat histopathology. The results showed that body weight increased in the control (G_0_, non-diabetic) and decreased in the positive control (G_1_, diabetic). The reduction in body weight in diabetic rats might show abnormalities in glucose lipid metabolism, leading to tissue breakdown and muscle weakness. These results are consistent with [13], as they documented a reduction in body weight of diabetic rats from 165.93 ± 1.41 to 143.75 ± 2.20 after the 14th week of diabetes induction in rats. Further, these results also align with [14], where methanol extract of strawberry was orally administered to diabetes-induced rats, with documented initial means of 175 ± 1.2 that were increased to 185.16 ± 3.79 g at trial termination. Additionally, the literature suggests that a decrease in hyperglycemia level reduced the risk of cardiovascular complications and our results found that strawberry extract affected these parameters in rats. Our results are also in agreement with [15], in which they found the effect of strawberry on diabetic patients, and their data showed that blood glucose levels decreased from 172.09 ± 28.648 to 141.00 ± 38.079 after 14 days of strawberry (200 g/person/day) consumption. A similar result was also reported by [16], in which they explained that in vitro aqueous extracts of strawberry fruits inhibited enzyme (α-amylase and α-glucosidase) activities that control blood sugar levels. The strawberry extract showed a significant increase in plasma insulin levels from the diabetic control to the strawberry-extract-treatment group, since the liver releases carbohydrates and fatty acids from storage that play a crucial role in metabolism [17]. A biomarker of liver bilirubin plays a defensive role in both cardiovascular diseases (CVD) as well as in metabolism disorders by acting as an antioxidant and cytoprotective factor, and it was correlated with levels of diabetes mellitus in a study conducted by [18]. In the strawberry-extract-treated group, serum bilirubin reverted to normal levels. The results of the present findings are in agreement with [19], who also measured the serum bilirubin level of 1342 patients with T2DM. They documented the mean of diabetic patients to be 9.96 ± 3.60, which was close to the normal value of 11.63 ± 3.87. Liver enzymes, alanine aminotransferases (ALT), aspartate aminotransferase (AST) and alkaline phosphatase also revert to normal values, and our study findings aligned with [16], where a liver function test was performed in a diabetic population, and the findings revealed that ALT was elevated in 40.4% of the diabetic population, while AST and ALP levels were elevated in 17% and 16% of diabetic population, respectively. It was concluded that patients who have type 2 diabetes mellitus have a higher likelihood of having abnormal liver function tests than those without diabetes. Our results of plasma total protein, albumin, globulin levels and albumin/globulin ratio (A/G) are also in agreement with them. The findings of antioxidant enzymes are consistent with the report of [20], who documented SOD and GPX values of 2.08 ± 0.03 and 31.04 ± 0.97, respectively, in the control, and the values were decreased (0.36 ± 0.04 and 8.83 ± 1.61) in the diabetic group as well as increased (1.83 ± 0.03 and 28.80 ± 3.38) in the treatment group. The studies showed that changes in antioxidant enzyme levels make tissues more susceptible to oxidative stress, resulting in diabetic complications. The results of the lipid profile findings are in agreement with [16], who showed that in the control groups, values of HDL, LDL, VLDL and TG were 28.73 ± 3.25, 27.4 ± 2.29¸ 6.13 ± 0.46 and 30.62 ± 2.30, and these values increased in the diabetic groups to 55.99 ± 6.33, 138.65 ± 7.70, 19.01 ± 0.34 and 95.12 ± 1.69, respectively. Diabetes affects the pancreas, as indicated by performed histopathology examination. The literature suggested that alloxan, for instance, kills the β cells of the pancreas, and interventions can also be detected through histological examination the cell tissues [21]. Our results showed that using alloxan monohydrate as a single dose damaged the pancreatic cells in rats, while rats receiving strawberry fruit extract at 500 mg/kg for 28 days showed near normal islets of Langerhans and β cells of the rat pancreas.

## 5. Conclusions

In conclusion, significant differences (*p* < 0.05) were found between diabetic groups that were treated with strawberry fruit extract and the positive control (diabetic, normal diet) group in weight, blood glucose, LFT, RFT, lipid profile, and antioxidant parameter levels. The strawberry fruit extract showed positive results against these diabetic parameters and reduced the severity of hyperglycemia by enhancing plasma insulin levels. It also enhanced the antioxidant parameters of GPx and SOD levels with a level of strawberry fruit extract at 500 mg/kg body weight. Considering the findings of this study, strawberry fruit extract consumption is a pragmatic approach to manage hyperglycemic and hyperlipidemic levels in diabetes patients.

## Figures and Tables

**Figure 1 foods-12-02911-f001:**
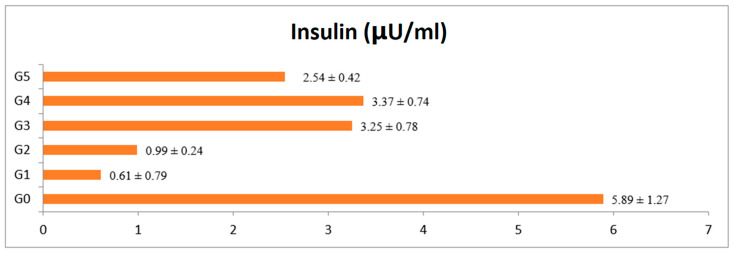
Effect of strawberry fruit extract on serum insulin level of rats. All data expressed as mean ± SD. G_0_ (negative Control), G_1_ (Positive control), G_2_, G_3_ and G_4_ given 250, 500, 750 mg/kg of BW strawberry extract respectively and G_5_ group given 70 mg/kg BW of metformin.

**Figure 2 foods-12-02911-f002:**
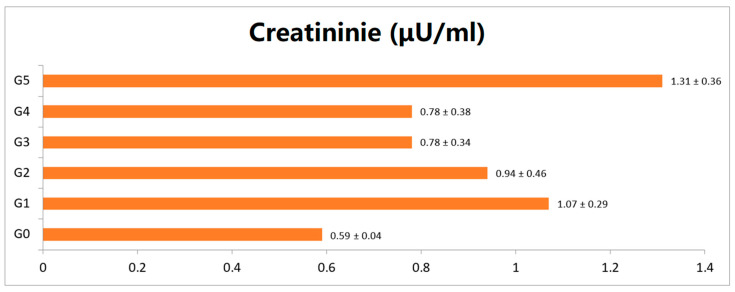
Effect of strawberry fruit extract on serum creatinine level (All data expressed as mean ± SD. G_0_ (negative control), G_1_ (positive control), G_2_, G_3_ and G_4_ given 250, 500, 750 mg/kg of BW strawberry extract, respectively, and G_5_ group given 70 mg/kg BW of metformin.

**Figure 3 foods-12-02911-f003:**
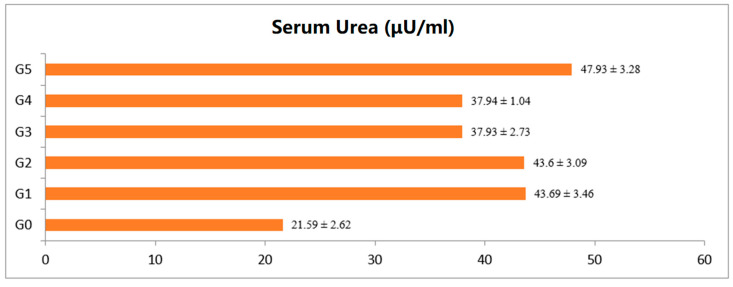
Serum urea level of rats provided with different levels of strawberry fruit extract. All data expressed as mean ± SD. G_0_ (negative control), G_1_ (positive control), G_2_, G_3_ and G_4_ given 250, 500, 750 mg/kg of BW strawberry extract, respectively, and G_5_ group given 70 mg/kg BW of metformin.

**Figure 4 foods-12-02911-f004:**
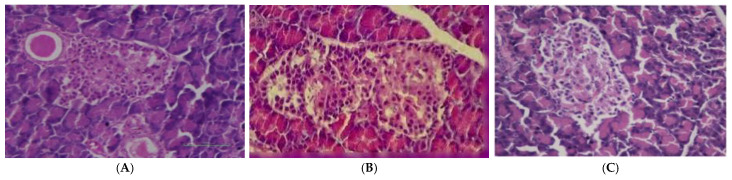
Histopathological changes in the pancreas of rats provided with strawberry fruit extract, (**A**) Control group (almost normal; no tissue changes were seen); (**B**) 1st day of diabetes (destruction of pancreatic β cells); (**C**) 28th day of diabetes induction given strawberry extract (SE) at 500 mg/kg BW (restoration of pancreatic β cells).

**Table 1 foods-12-02911-t001:** Effect of strawberry fruit extract provision on body weight and blood glucose of rats.

Groups	Body Weight		Serum Glucose	
	Baseline	28th Day	Baseline	28th Day
G_0_	155.04 ^B^ ± 7.44	205.89 ^aA^ ± 7.56	87.23 ^a^ ± 3.06	88.3 ^e^ ± 4.50
G_1_	158.42 ^A^ ± 6.53	137.51 ^efB^ ± 4.37	409.65 ^bB^ ± 18.87	474.85 ^aA^ ± 20.01
G_2_	154.03 ^A^ ± 7.54	142.8 ^deB^ ± 2.56	413.25 ^bA^ ± 17.35	369.21 ^bB^ ± 17.02
G_3_	157.01 ± 7.23	148.84 ^cd^ ± 4.56	392.53 ^bA^ ± 20.26	197.48 ^dB^ ± 13.52
G_4_	156.04 ± 7.42	147.7 ^b^ ± 3.15	402.41 ^bA^ ± 19.51	231.41 ^cdB^ ± 9.22
G_5_	155.04 ^A^ ± 7.34	138.43 ^fB^ ± 3.61	397.34 ^bA^ ± 16.03	269.97 ^cB^ ± 14.53

All data expressed as mean ± SD. Means superscripted with different letters are significantly different (*p* < 0.05); G_0_ (negative control), G_1_ (positive control), G_2_, G_3_ and G_4_ given 250, 500, 750 mg/kg BW strawberry extract, respectively, and G_5_ group given 70 mg/kg BW of metformin.

**Table 2 foods-12-02911-t002:** Liver function test of rats provided with strawberry fruit extract for a period of 28 days.

Liver Function Test (LFT) Parameters (mg/dL)	Control Groups		Diabetic Groups			
	G_0_	G_1_	G_2_	G_3_	G_4_	G_5_
Bilirubin Total	0.24 ± 0.02 ^a^	0.16 ± 0.03 ^d^	0.18 ± 0.01 ^b^	0.20 ± 0.04 ^bc^	0.19 ± 0.02 ^bc^	0.19 ± 0.06 ^bc^
A.L.T (S.G.P.T)	43.31 ± 1.23 ^c^	56.44 ± 1.04 ^a^	50.32 ± 0.99 ^b^	45.96 ± 0.75 ^c^	46.11 ± 0.76 ^c^	51.45 ± 1.19 ^b^
A.S.T (S.G.O.T)	74.50 ± 1.98 ^d^	219.71 ± 0.78 ^a^	212.50 ± 1.20 ^b^	199.47 ± 1.90 ^c^	199.47 ± 1.90 ^c^	214.22 ± 0.99 ^b^
Alkaline Phosphatase	152.00 ± 7.81 ^e^	347.33 ± 4.51 ^a^	186.67 ± 2.08 ^b^	164.54 ± 1.94 ^d^	164.53 ± 2.04 ^d^	246.00 ± 4.36 ^c^
Protein Total *	6.51 ± 0.01 ^a^	6.41 ± 0.01 ^c^	6.45 ± 0.03 ^b^	6.52 ± 0.02 ^a^	6.52 ± 0.01 ^a^	6.45 ± 0.01 ^b^
Albumin *	3.42 ± 0.02 ^a^	3.00 ± 0.02 ^c^	3.18 ± 0.02 ^b^	3.41 ± 0.01 ^a^	3.43 ± 0.02 ^b^	3.19 ± 0.05 ^b^
Globulins *	1.28 ± 0.77 ^c^	1.23 ± 0.25 ^c^	1.31 ± 0.56 ^b^	1.32 ± 0.58 ^b^	1.42 ± 0.29 ^a^	1.25 ± 0.13 ^c^
A/G Ratio	2.67 ± 0.01 ^a^	2.44 ± 0.13 ^c^	2.42 ± 0.31 ^c^	2.58 ± 0.04 ^b^	2.41 ± 0.03 ^c^	2.55 ± 0.07 ^b^

All data expressed as mean ± SD. Means superscripted with different letters are significantly different (*p* < 0.05). ALT, alanine transaminase; SGPT, serum glutamic pyruvic transaminase; AST, aspartate aminotransferase, SGOT, serum glutamic–oxaloacetic transaminase. G_0_ (negative control), G_1_ (positive control), G_2_, G_3_ and G_4_ given 250, 500, 750 mg/kg BW strawberry extract, respectively, and G_5_ group given 70 mg/kg BW of metformin. * values described in g/dL.

**Table 3 foods-12-02911-t003:** Effect of strawberry fruit extract on antioxidant parameters of rats.

Parameters (U/mL)	Groups					
	Control		Diabetic			
	G_0_	G_1_	G_2_	G_3_	G_4_	G_5_
Glutathione Peroxidase (GPx)	2.83 ± 0.21 ^a^	0.82 ± 0.21 ^e^	0.95 ± 0.01 ^c^	1.54 ± 0.21 ^b^	1.72 ± 0.34 ^b^	0.87 ± 0.36 ^d^
Superoxide dismutase (SOD)	7.53 ± 0.31 ^a^	2.54 ± 0.26 ^e^	4.13 ± 0.18 ^c^	4.74 ± 0.23 ^c^	5.49 ± 0.32 ^b^	2.72 ± 0.41 ^d^

All data expressed as mean ± SD. Means superscripted with different letters are significantly different (*p* < 0.05). G_0_ (negative control), G_1_ (positive control), G_2_, G_3_ and G_4_ given 250, 500, 750 mg/kg BW strawberry extract, respectively, and G_5_ group given 70 mg/kg BW of metformin.

**Table 4 foods-12-02911-t004:** Effect of strawberry fruit extracts levels on plasma lipid profile of rats.

Parameters (mg/dL)	Control		Diabetic Groups			
	G_0_	G_1_	G_2_	G_3_	G_4_	G_5_
Serum *Total cholesterol*	71.00 ± 1.00 ^bc^	75.21 ± 1.02 ^a^	71.34 ± 1.48 ^bc^	69.45 ± 0.98 ^c^	69.24 ± 1.01 ^c^	72.25 ± 1.430 ^b^
Serum Triglycerides	103.33 ± 1.15 ^e^	135.33 ± 1.53 ^a^	121.33 ± 1.53 ^c^	115.33 ± 1.53 ^d^	116.33 ± 1.53 ^d^	126.67 ± 2.08 ^b^
Serum High-density lipoprotein cholesterol	24.24 ± 1.30 ^a^	15.21 ± 1.09 ^b^	17.23 ± 1.02 ^b^	23.13 ± 1.73 ^a^	23.67 ± 1.15 ^a^	18.03 ± 1.05 ^b^
Serum (Low-density lipoprotein cholesterol)	14.41 ± 1.2 ^c^	21.67 ± 0.54 ^a^	18.22 ± 1.42 ^b^	13.56 ± 1.41 ^c^	13.54 ± 1.32 ^c^	17.02 ± 1.65 ^b^
Serum (Non-high-density lipoprotein cholesterol)	46.45 ± 1.43 ^b^	51.63 ± 1.21 ^a^	51.24 ± 1.52 ^a^	43.42 ± 1.28 ^b^	43.36 ± 1.53 ^b^	51.33 ± 1.53 ^a^

All data expressed as mean ± SD. Means superscripted with different letters are significantly different (*p <* 0.05). TC, total cholesterol; TG = triglyceride; HDL = high-density lipoproteins; LDL = low-density lipoprotein. G_0_ (negative control), G_1_ (positive control), G_2_, G_3_ and G_4_ given 250, 500, 750 mg/kg of BW strawberry extract, respectively, and G_5_ group given 70 mg/kg BW of metformin.

## Data Availability

The study data and analysis are included in this publication.

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
