# Peer review of "Antidiabetic, Antioxidative and Antihyperlipidemic Effects of Strawberry Fruit Extract in Alloxan-Induced Diabetic Rats"

_foods, 2023, doi:10.3390/foods12152911_

Round 1

Reviewer 1 Report

The research article by Iftikhar Younis Mallhi et al. presented the work entitled Antidiabetic, Antioxidative and Antihyperlipidemic effect of strawberry fruit extract in Alloxan-Induced Diabetic Rats”. Authors have taken into consideration the use of the strawberry extract to study the three factors (Antidiabetic, Antioxidative and Antihyperlipidemic) of diabetes by using Alloxan-Induced diabetic rats. Phytochemicals are naturally occurring molecules found in plants, such as medicinal herbs, vegetables, and fruits that combine with nutrients and fibres to act against or particularly protect against illness. Plants have been a source of a vast array of biologically active substances for ages and have been employed widely as crude material or purified compounds to cure several diseases. Traditional plant-based medicine plays a vital part in the growth and advancement of current research by acting as a launching point for the creation of new drug discovery techniques. Various contemporary medications were derived from traditional medicinal plants using plant material, following ethnobotanical leads derived from indigenous treatments utilized by traditional medical systems. In underdeveloped nations and rural communities, medicinal plants are both a significant resource and a need, and it also offers a viable alternative to primary healthcare institutions. A substantial amount of editing is required for the current version of the article. The following ideas should be taken into account.

·       To enhance the first-sight quality of the research article, the abstract requires thorough rewriting to correct its language and grammar errors.

·       The article's title is not heavily discussed in the introduction. Make an effort to add more data and link information on antihyperlipidemic, antioxidation, and antidiabetes.

·       The language used in the experimental setup and data presentation is not easily understood.

·       The proper information is not provided in section 2.2. How much of the sample was taken, how was it dried, and how long was it dried for? Please improve this section since published articles are used as sources for future research.

·       Section 2.5. The language in this section needs to be revised; please revise it carefully and rectify all errors.

·       Line 112 “of”. Correct preposition.

·       Line 113 “refine” Correct the verb form.

·       The authors don't give any explanation for why they settled on the Chandler type of strawberries in terms of its intended use.

·       Line 128. Missing closing punctuation.

·       Line 130 “execution” “research”. Add article.

·       Line 134 “The group”. Consider removing article

·        Line 134 “also”. Missing verb.

·       Line 146. Missing closing punctuation.

·       Line 153. “After blood sampling serum was collected after centrifugation”. Consider this line for grammatical errors.

·       Section 2.7. Split the Gpx methodology data into the next paragraph.

·       Line 276. “Our Results also lien to [21] in which they measured the Serum bilirubin total level in 1342 patients with type 2 diabetes mellitus (T2DM) and documented the results mean value of normal patients is 11.63 ± 3.87 and mean value of diabetic patients is 9.96 ± 3.60”. Consider rewriting this sentence and spelling and grammartical errors.

·       Line 325. “Renal markers such as urea and showed that Group (G1) animals were recorded with increased values of these parameters in comparison with normal control group (G0) rats”. Why authors included this statement in this section when they have written results on renal functions test in another section?

·       Why authors took only these two renal markers in this study? Any specific reason?

·       Figure 4. Requires editing.

·       The discussion part in each section may be further refined to present a clear coherent defense of your findings.

·       The report indicates that the plagiarism percentage is 22%; you should strive to minimize this up to 15%.

Add the following important Reference

https://pubmed.ncbi.nlm.nih.gov/30532634/

Author Response

Reviewer 1

The research article by Iftikhar Younis Mallhi et al. presented the work entitled “Antidiabetic, Antioxidative and Antihyperlipidemic effect of strawberry fruit extract in Alloxan-Induced Diabetic Rats”. Authors have taken into consideration the use of the strawberry extract to study the three factors (Antidiabetic, Antioxidative and Antihyperlipidemic) of diabetes by using Alloxan-Induced diabetic rats. Phytochemicals are naturally occurring molecules found in plants, such as medicinal herbs, vegetables, and fruits that combine with nutrients and fibres to act against or particularly protect against illness. Plants have been a source of a vast array of biologically active substances for ages and have been employed widely as crude material or purified compounds to cure several diseases. Traditional plant-based medicine plays a vital part in the growth and advancement of current research by acting as a launching point for the creation of new drug discovery techniques. Various contemporary medications were derived from traditional medicinal plants using plant material, following ethnobotanical leads derived from indigenous treatments utilized by traditional medical systems. In underdeveloped nations and rural communities, medicinal plants are both a significant resource and a need, and it also offers a viable alternative to primary healthcare institutions. A substantial amount of editing is required for the current version of the article. The following ideas should be taken into account.

To enhance the first-sight quality.

Q1. To enhance the first-sight quality of the research article, the abstract requires thorough rewriting to correct its language and grammar errors.

Ans: Thank you so much for your suggestions. The abstract has been revised as per suggestion and grammatical errors have been corrected.

Q2. The article's title is not heavily discussed in the introduction. Make an effort to add more data and link information on antihyperlipidemic, antioxidants, and antidiabetes.

Ans: Thank you very much for comments. Introduction has been revised as per suggestion of the reviewers.

Q3. The language used in the experimental setup and data presentation is not easily understood.

Ans: Thank you for your suggestion, the language of experiments and data interpretation is revised and explained

Q4. The proper information is not provided in section 2.2. How much of the sample was taken, how was it dried, and how long was it dried for? Please improve this section since published articles are used as sources for future research.

Ans: Thank you so much for your valuable suggestion. The section 2.2 has been improved and explained as per suggestion of the reviewer.

Q5. Section 2.5.The language in this section needs to be revised; please revise it carefully and rectify all errors

Ans: thank you so much section 2.5 revised

Q6. Line 112 “of”. Correct preposition.

Ans: Done, thank you for correcting

Q7. Line 113 “refine” Correct the verb form.

Ans: The word, “refine” change into refined

Q8. The authors don't give any explanation for why they settled on the Chandler type of strawberries in terms of its intended use.

Ans: Thank you so much, I have removed the word Chandler and explained that local variety was used along with its intended use.

Q9. Line 128: Missing closing punctuation

Ans: Closing punctuation added, thank you

Q10. Line 130 “execution” “research”. Add article.

Ans: thank you so much, article has been added

Q11. Line 134 “The group”. Consider removing article

Ans: Thank you so much, corrected by the word “groups” removed

Q12. Line 134 “also”. Missing verb.

Ans: Thank you so much, corrected

 Q13. Line 146. Missing closing punctuation

Ans. Thank you so much, added

Q14.Line 153. “After blood sampling serum was collected after centrifugation”. Consider this line for grammatical errors.

Ans: corrected, thank you so much for your suggestion

Q15. Section 2.7. Split the Gpx methodology data into the next paragraph.

Ans: Gpx methodology data added into the next paragraph, thank you for your suggestion

Q16.  Line 276. “Our Results also lien to [21] in which they measured the Serum bilirubin total level in 1342 patients with type 2 diabetes mellitus (T2DM) and documented the results mean value of normal patients is 11.63 ± 3.87 and mean value of diabetic patients is 9.96 ± 3.60”. Consider rewriting this sentence and spelling and grammatical errors.

Ans: Thank you so much for your suggestion, this sentence revised

Q17. Line 325. “Renal markers such as urea and showed that Group (G1) animals were recorded with increased values of these parameters in comparison with normal control group (G0) rats”. Why authors included this statement in this section when they have written results on renal functions test in another section?

Ans: corrected, thank you so much for valuable suggestion.

 Q18. Why authors took only these two renal markers in this study? Any specific reason?

Ans: thank you, although there are several biomarkers that can be considered but these are main biomarkers for diabetes millatus so they were included in the study  

Q19.  Figure 4. Requires editing

Ans: Thank you so much, figure 4 has been edited as per suggestions.

Q20. The discussion part in each section may be further refined to present a clear coherent defense of your findings.

Ans: In accordance with journal formats, the Discussion and Results sections have been separated and refined as well as explained in more detail.

Q21. The report indicates that the plagiarism percentage is 22%; you should strive to minimize this up to 15%.

Ans: thank you so much, the plagiarism percentage minimized (10%) as per the directions of the reviewer

Q 22. Add the following important Reference https://pubmed.ncbi.nlm.nih.gov/30532634/

Ans. The suggested reference has been cited in the manuscript.

Reviewer 2 Report

In this manuscrpit, the author describes the potential of strawberry fuit extract in antidiabetic, antioxidant and antihyprlipidemic activity on alloxan-iduced diabetic rats. The use of this natural raw material for applications is very well-established and many articels have been published in recent years in similar thematic areas. Hence, in principle this review does not add any novelty and does not introduce significant content for the scientific community.

However, despite the lacking any element of novelty, the authors tried to nicely organize the different types of activity, and this is remarkable since they belong to a wide group of prohealt acting with distinct effects and properties. The readers could benefit from this classification. For this reason  I strongly encourage the authors to consider significant changes which should be introduced. With these changes, the manuscript will get the improving that actually lacks.

 In particular, the authors should take in mind the following suggestion:

 All presented and analyzed activities are strongly dependent on the presence of biologically active substances in the tested material. Strawberry fruits are rich in various groups of polyphenolic compounds (anthocyanins, ellagitannins, etc.). Many scientific studies over the last decades have indicated the dependence between the presence of specific classes of polyphenols as well as their concentration in the tested material. Therefore, I believe that chemical characterization of the tested plant material is necessary. I suggest performing spectrophotometric tests for the content of polyphenols in total and their individual classes. In my opinion, this is a crucial for the manuscript.

Author Response

Reviewer 2

  1. In this manuscrpit, the author describes the potential of strawberry fruit extract in antidiabetic, antioxidant and antihyprlipidemic activity on alloxan-iduced diabetic rats. The use of this natural raw material for applications is very well-established and many articels have been published in recent years in similar thematic areas. Hence, in principle this review does not add any novelty and does not introduce significant content for the scientific community. However, despite the lacking any element of novelty, the authors tried to nicely organize the different types of activity, and this is remarkable since they belong to a wide group of ProHealth acting with distinct effects and properties. The readers could benefit from this classification. For this reason, I strongly encourage the authors to consider significant changes which should be introduced. With these changes, the manuscript will get the improving that actually lacks.

Ans. Thank you very much for your comments. We appreciate the efforts of the reviewer for critically reviewing the manuscript and pointing out changes.  

 In particular, the authors should take in mind the following suggestion: All presented and analyzed activities are strongly dependent on the presence of biologically active substances in the tested material. Strawberry fruits are rich in various groups of polyphenolic compounds (anthocyanins, ellagitannins, etc.). Many scientific studies over the last decades have indicated the dependence between the presence of specific classes of polyphenols as well as their concentration in the tested material. Therefore, I believe that chemical characterization of the tested plant material is necessary. I suggest performing spectrophotometric tests for the content of polyphenols in total and their individual classes. In my opinion, this is a crucial for the manuscript.

Ans. The reviewer has rightly pointed out the characterization of strawberry fruit with particular reference to the specific classes of polyphenols as well as their concentration in tested material. We have characterized the strawberry fruit samples before starting the biological efficacy of the study and measured the total phenolics, total flavonoid and anthocyanin content of all the local varieties of the strawberry fruit samples and can be added in the manuscript if required.

Round 2

Reviewer 1 Report

The authors have revised the manuscript as per suggestions.  

Accept

Author Response

Dear Editor in Chief,

Foods Journal

First of all, thank you very much for sending the comments of the reviewers. Please find the revised manuscript titled “Antidiabetic, Antioxidative and Antihyperlipidemic effects of strawberry fruit extract in Alloxan-Induced Diabetic Ratswhich is original research conducted in the Department of Food Science and Human Nutrition, University of Veterinary and Animal Sciences, Lahore Pakistan and have been prepared for submission in “Foods”. As per reviewer’s comments and suggestions, the changes have been made in below mentioned comments with detail as below

  1. The authors have revised the manuscript as per suggestions. Accept

Ans. Thank you very much for the efforts of the reviewer to evaluate the paper and the comments

Reviewer 2 Report

If the authors declare to perform screening analyzes in terms of the assessment of the content of polyphenolic compounds and they are ready, I suggest placing them as supplemantary materials.

Author Response

Dear Editor in Chief,

Foods Journal

First of all, thank you very much for sending comments of the reviewers. Please find the revised manuscript titled “Antidiabetic, Antioxidative and Antihyperlipidemic effects of strawberry fruit extract in Alloxan-Induced Diabetic Ratswhich is original research conducted in the Department of Food Science and Human Nutrition, University of Veterinary and Animal Sciences, Lahore Pakistan. As per reviewer’s comments and suggestions, the changes have been made in below mentioned comments with detail as below

Reviewer 2

Q: If the authors declare to perform screening analyzes in terms of the assessment of the content of polyphenolic compounds and they are ready, I suggest placing them as supplemantary materials.

Ans: Thank you very much for the comments. We have analyzed total phenolic (TPC), total flavonoids (TFC) and Anthocyanins (AC) contents of different strawberry varieties (festival, chandler and korona) produced in Pakistan before starting the trial and based on the nutritional indicators, phytochemicals and antioxidants analysis. Chandler variety was used for the bioefficacy trial. The results for the total phenolic (TPC), total flavonoids (TFC) and Anthocyanins (AC) contents for the supplementary material is as below

Table. Total phenolic, total flavonoid and anthocyanin ccontent of different strawberry varieties of Pakistan

Parameters

Varieties

Mean ± SD

Total phenolic Contents

(mg GAE/100 g)

Chandler

926.17 ± 53.08a

Korona

871.55 ± 48.48b

Festival

864.44 ± 44.16b

Total flavonoids Contents

(mg/100g)

Chandler

180.63 ± 12.72a

Korona

144.84 ± 10.11c

Festival

163.62 ± 11.42b

Anthocyanins

(mg/100g)

Chandler

2.74 ± 0.09a

Korona

1.50 ± 0.04c

Festival

1.98 ± 0.05b

Data are expressed as “mean ± standard deviation”. Superscript letters (a, b) shows the statistically differences. Different letters indicate significant differences (P < 0.05, Tukey test) between strawberry varieties.